# Simultaneous Quantitative Detection of HCN and $C_2H_2$ in Combustion Environment Using TDLAS

**Wubin Weng *** , **Marcus Aldén and Zhongshan Li**

Division of Combustion Physics, Lund University, P.O. Box 118, SE-221 00 Lund, Sweden;
marcus.alden@forbrf.lth.se (M.A.); zhongshan.li@forbrf.lth.se (Z.L.)
* Correspondence: wubin.weng@forbrf.lth.se; Tel.: +46-46-222-3208

**Abstract:** Emission of nitrogen oxides ($NO_x$) and soot particles during the combustion of biomass fuels and municipal solid waste is a major environmental issue. Hydrogen cyanide (HCN) and acetylene ($C_2H_2$) are important precursors of $NO_x$ and soot particles, respectively. In the current work, infrared tunable diode laser absorption spectroscopy (IR-TDLAS), as a non-intrusive in situ technique, was applied to quantitatively measure HCN and $C_2H_2$ in a combustion environment. The P(11e) line of the first overtone vibrational band $v_1$ of HCN at 6484.78 $cm^{-1}$ and the P(27e) line of the $v_1 + v_3$ combination band of $C_2H_2$ at 6484.03 $cm^{-1}$ were selected. However, the infrared absorption of the ubiquitous water vapor in the combustion environment brings great uncertainty to the measurement. To obtain accurate temperature-dependent water spectra between 6483.8 and 6485.8 $cm^{-1}$, a homogenous hot gas environment with controllable temperatures varying from 1100 to 1950 K provided by a laminar flame was employed to perform systematic IR-TDLAS measurements. By fitting the obtained water spectra, water interference to the HCN and $C_2H_2$ measurement was sufficiently mitigated and the concentrations of HCN and $C_2H_2$ were obtained. The technique was applied to simultaneously measure the temporally resolved release of HCN and $C_2H_2$ over burning nylon 66 strips in a hot oxidizing environment of 1790 K.

**Keywords:** hydrogen cyanide; acetylene; tunable diode laser absorption spectroscopy; combustion/gasification; hot water interference; biomass/waste

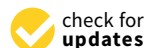

## 1. Introduction

The combustion of biomass fuels and municipal solid waste is an important fossil-free energy sector providing heat and power. However, the emission of nitrogen oxides ($NO_x$) and soot particles during combustion is seen as a major environmental issue. Hydrogen cyanide (HCN) and acetylene ($C_2H_2$) are important precursors of $NO_x$ and soot particles, respectively. To fully understand the $NO_x$ and soot particle formation process in various combustion environments, reliable measurements of HCN and $C_2H_2$ are essential. Considering that both HCN and $C_2H_2$ will be consumed downstream, nonintrusive in situ laser-based techniques are preferred.

Due to the lack of proper electronic transitions in the UV/visible spectral regions, the detection of HCN in a combustion environment is mainly focused on using the absorption lines in the infrared region. Sun et al. [1] demonstrated quantitative measurements of HCN in premixed $CH_4/N_2O/O_2/N_2$ flames using midinfrared polarization spectroscopy (IRPS). The P20 line of the fundamental C–H stretching band at around 3248 $cm^{-1}$ was selected. Using the same absorption line, Hot et al. [2] achieved quantitative in situ measurements of HCN released from burning straw pellets at atmospheric pressure using mid-infrared degenerate four-wave mixing (IR-DFWM). Goldman et al. [3] reported HCN measurement in low-pressure flames using fiber laser intracavity absorption spectroscopy (FLICAS), probing the first overtone vibrational band at around 1.5 μm. In addition, ex situ measurements of HCN in flames were reported by Gersen et al. [4], using wavelength

modulation absorption spectroscopy (WMAS) to probe the P(13) line of the first overtone, and Lamoureux et al., using pulsed cavity ring-down spectroscopy (CRDS) [5] and continuous-wave CRDS [6] to probe the second and first overtone. Among the different available techniques, tunable diode laser absorption spectroscopy (TDLAS) is competitive, which is relatively less complex, more robust, and cost effective compared to the others. In TDLAS measurement, a diode laser operates in a single longitudinal mode and provides single-frequency emission with a narrow linewidth. The laser emission scans in wavelength to resolve the atom/molecule absorption lines, and the atom/molecule concentration is derived based on the Beer–Lambert law, well-known to measure various atoms [7] and molecules [8,9] with high detection sensitivity and precision. In the present work, TDLAS was developed for quantitative in situ measurement of HCN in a harsh biomass/waste combustion environment by probing the first overtone ro-vibrational band of HCN at 1.5 μm (Figure 1).

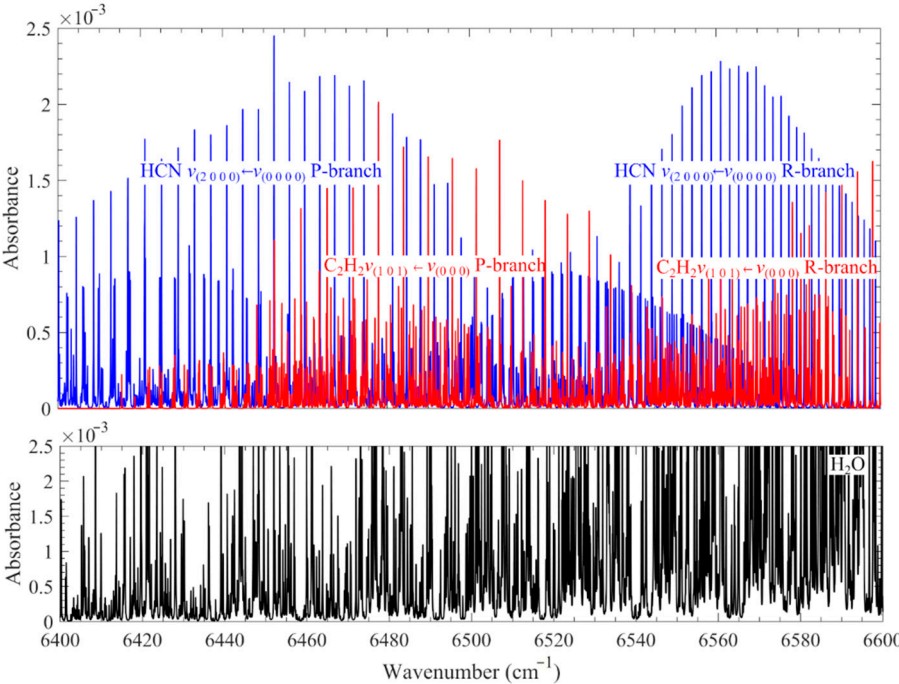

**Figure 1.** Calculated absorbance of 1000 ppm HCN, 1000 ppm $C_2H_2$, and 10% $H_2O$ over 6400–6600 $cm^{-1}$ at 1200 K and atmospheric pressure with an optical path length of 0.34 m. The spectra data from the HITRAN2016 database were adopted for spectral simulation.

Compared to HCN, more laser diagnostics have been developed for quantitative measurements of $C_2H_2$ in flames. The techniques include IRPS [10,11], coherent anti-Stokes Raman scattering (CARS) [12,13], laser-induced fluorescence (LIF) [14], spontaneous Raman scattering (SRS) [15–18], and tunable diode laser absorption spectroscopy (TDLAS) [19,20]. In a harsh combustion environment, the strong broadband interference from the laser-induced fluorescence of polycyclic aromatic hydrocarbon (PAH) and the laser-induced incandescence (LII) from soot particles suppresses the weak signal of $C_2H_2$ LIF and spontaneous Raman scattering. Recently, Kim et al. [18] developed time-resolved polarization lock-in filtering for background suppression. Techniques based on the probing of the IR absorption line seem to be more viable. To manage the simultaneous measurement of HCN, the $v_1 + v_3$ vibrational combination band at 1.5 μm (see Figure 1) was selected.

However, in combustion environments, challenges arise from the strong absorption of hot water lines at 1.5 μm, interfering with the HCN and $C_2H_2$ absorption signal, as shown in Figure 1. To minimize line interference, it is necessary to select suitable HCN and $C_2H_2$ lines.

Figure 2 shows the high-resolution spectra of 1000 ppm HCN, $C_2H_2$, and 10% $H_2O$ over 6481–6489 cm$^{-1}$ at 1200 K and atmospheric pressure with an optical path length of 0.34 m. The calculation was carried out using the HITRAN2016 [21] and HITEMP2010 [22] databases. The interference from other typical species, such as CO, $CO_2$, $C_2H_4$, and $NH_3$, that would be released from burning biomass/waste combustion was also considered. Among these species, $NH_3$ has the strongest absorption at this wavelength, and the absorption spectrum of 1000 ppm $NH_3$ is also presented in Figure 2. In the present work, the P(11e) line of the first overtone vibrational band $v_1$ of HCN at 6484.78 cm$^{-1}$ with a line intensity ($S$) of $5.639 \times 10^{-21}$ cm/molecule at 296 K was selected, and the typical absorption curve of this line is presented in the inset of Figure 2. At the same time, a nearby $C_2H_2$ absorption line, which is the P(27e) line of the $v_1 + v_3$ combination band at 6484.03 cm$^{-1}$ with a line intensity ($S$) of $8.113 \times 10^{-22}$ cm/molecule at 296 K, was used for $C_2H_2$ measurement.

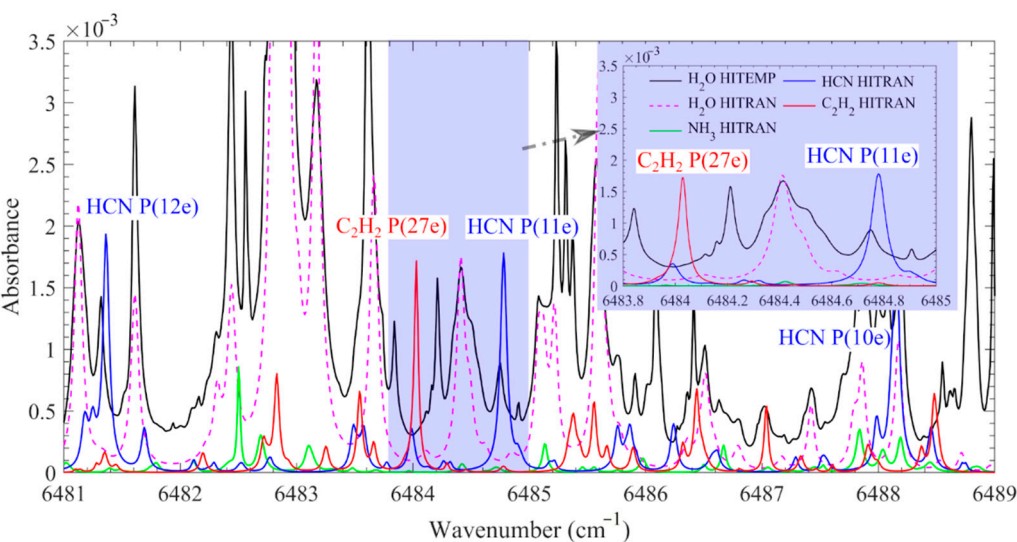

**Figure 2.** Calculated absorbance of 1000 ppm HCN, $C_2H_2$, $NH_3$, and 10% $H_2O$ over 6481–6489 cm$^{-1}$ at 1200 K and atmospheric pressure with an optical path length of 0.34 m. The spectra data from the HITRAN2016 and HITEMP2010 databases were used for spectral simulation. Spectra with high-resolution at 6483.8–6485 cm$^{-1}$ are shown in the inset.

However, using the selected HCN and $C_2H_2$ lines, it is still impossible to completely avoid the interference of the hot water lines, as shown in the inset in Figure 2. Moreover, the HITRAN2016 and HITEMP2010 databases present different water absorption spectra. To further minimize the interference with the HCN and $C_2H_2$ measurement, a systematic investigation of the water line in the 6483.8–6485 cm$^{-1}$ spectral range is needed. In the present work, a homogenous hot gas environment provided by a laminar flame burner covering temperatures from 1120 to 1950 K was employed to obtain the temperature-dependent, hot-water-line absorption spectra. The accurate water absorption data were used to mitigate the interference from the hot water lines, and the time-resolved concentration of HCN and $C_2H_2$ at 5 mm above burning nylon 66 strips was quantitatively and simultaneously measured.

## 2. Experimental Setup

The temperature-dependent absorption spectra of water vapor were collected in hot gas environments with temperatures ranging from 1120 to 1950 K. The hot gas environments were provided by laminar flames anchored on a multijet burner. The details of the burner were described by Weng et al. [23]. As shown in Figure 3a, the burner comprises of a jet chamber, a co-flow chamber, and a burner head. A methane–air–oxygen mixture was introduced into the jet chamber and evenly distributed into 181 jet tubes. At the outlet of the jet tubes, Bunsen-type premixed flames were stabilized to generate hot flue gas

containing hot water vapor. At the same time, a mixture of nitrogen and air used as the co-flow passed through the co-flow chamber and finally mixed with the hot gas product from the jet flames. After this mixing, a hot gas environment with a high degree of uniformity and a size of about 85 × 45 mm was generated in the burner head region, as shown in Figure 3a. The temperature of the hot gas can be adjusted from 1120 to 1950 K using the flame conditions listed in Table 1. The temperature at 5 mm above the burner outlet, where hot water absorption was investigated, was measured using two-line atomic fluorescence (TLAF) thermometry with indium atom. The details of the temperature measurement technique were described by Borggren et al. [24]. The temperature was evenly distributed in a region of about 70 × 40 mm', with a narrow transition zone on the edge [23,24]. The water concentration in different hot gas environments was obtained based on chemical equilibrium calculation. Five mass flow controllers (BRONKHORST HIGH-TECH BV, Ruurlo, Netherlands) were used to control the gas flow rate. A flow stabilizer was placed 35 mm above the burner outlet for flow stabilization. Moreover, the hot gas environment at 1790 K provided by flame F7 was used for burning nylon 66 strips. As shown in Figure 3c, 500 mg nylon 66 strips were carried by two ceramic rods and placed in the center of the hot flue gas.

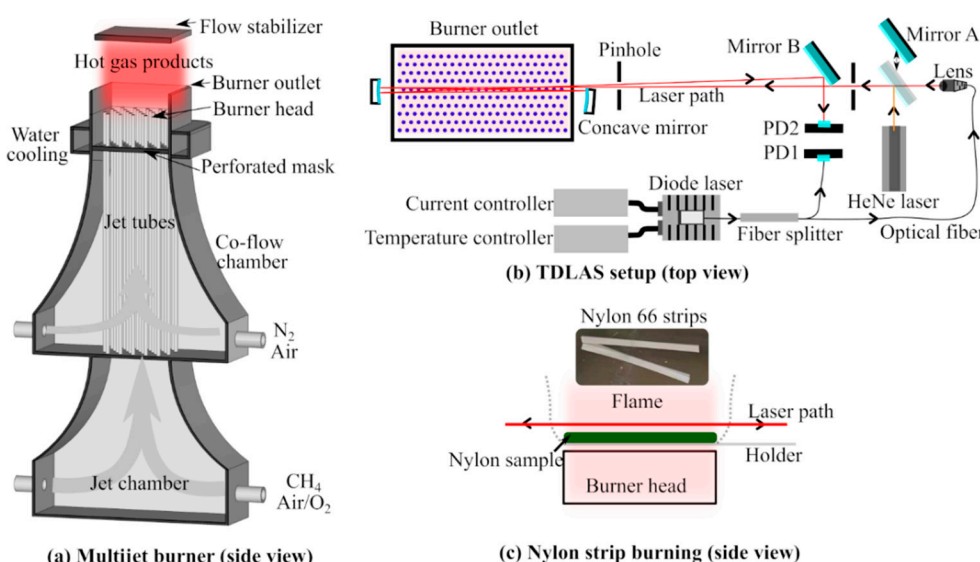

**Figure 3.** Schematic of the multijet burner (**a**), the TDLAS setup (**b**), and the measurement of HCN and $C_2H_2$ above burning nylon 66 strips (**c**). PD: photodiode. (Reproduced from Weng et al. [25] Copyright 2020 Elsevier.)

**Table 1.** Summary of the flame conditions, with corresponding temperature measured at 5 mm above the burner outlet and water concentration in the gas product.

| Flame Case | Gas Flow Rate (SLM) | | | | | Fuel/$O_2$ Equivalence Ratio φ | Gas Product Temperature (K) | $H_2O$ in Gas Product (%) |
| --- | --- | --- | --- | --- | --- | --- | --- | --- |
| | Jet-Flow | | | Co-Flow | | | | |
| | $CH_4$ | Air | $O_2$ | $N_2$ | Air | | | |
| F1 | 2.95 | 19.20 | 2.09 | 6.84 | 7.09 | 0.78 | 1950 | 15 |
| F2 | 2.66 | 17.34 | 1.89 | 10.83 | 7.74 | 0.74 | 1750 | 13 |
| F3 | 2.47 | 12.23 | 2.58 | 18.97 | 8.90 | 0.70 | 1550 | 11 |
| F4 | 2.28 | 11.89 | 2.26 | 22.69 | 9.83 | 0.67 | 1390 | 9 |
| F5 | 2.09 | 10.90 | 2.07 | 26.50 | 10.66 | 0.63 | 1260 | 8 |
| F6 | 1.71 | 8.91 | 1.69 | 26.92 | 10.25 | 0.60 | 1120 | 7 |
| F7 | 2.66 | 17.34 | 1.89 | 18.60 | 0.00 | 0.96 | 1790 | 13 |

A schematic of the optical setup of the TDLAS system is presented in Figure 3b. The laser beam was produced by a distributed feedback (DFB) diode laser (Butterfly, Toptica)

controlled by a combined laser diode and TEC controller (ITC4001, Thorlabs, LD current 1 A). The laser operates at temperatures of 5–45 °C to set it at a wavelength range of 1547–1551 nm and power of approximately 40 mW. In the presented work, the operating temperature was set to 36.4 °C and the laser wavelength at around 6485 $cm^{-1}$. By ramping up the driving current to between 0.15 and 0.25 A, the laser wavelength scanned over 2 $cm^{-1}$ at 100 Hz. The linewidth of the laser emission was below 1 MHz ($3.3 \times 10^{-4}$ $cm^{-1}$), which could well-resolve the absorption lines, such as water lines that have a FWHM of about 0.072 $cm^{-1}$ in a combustion environment. Using a fiber splitter, the laser was split into two parts. The one with 25% of the initial power was used as the reference beam, and the remaining portion was used for measurement, which efficiently eliminated the measurement uncertainty caused by laser energy fluctuation. The power of the reference beam was monitored by an IR photodiode (InGaAs, detecting wavelength 0.9–2.6 μm, PDA10D2, Thorlabs), i.e., PD1 in Figure 3b. The measurement beam was guided through the hot gas 5 mm above the burner outlet, or 5 mm above the burning nylon 66 strips (see Figure 3c), four times using two silver-coated concave mirrors ($f$ = 100 mm, $D$ = 50 mm, Thorlabs) to achieve a total optical path length of about 34 cm. The laser was finally collected by another IR photodiode (InGaAs, detecting wavelength 1.2–2.6 μm, PDA10D-EC, Thorlabs), i.e., PD2. A 632.8 nm laser beam provided by a HeNe laser was used as the alignment beam. Using the signal from PD1 and PD2, the number density ($N$) of HCN and $C_2H_2$ was derived based on Beer–Lambert law:

$$Abs = -\ln\left(\frac{(I(v) - I_s)/Tr}{I_0(v) - I_{s0}}\right) = S(T) \cdot g(v - v_0) \cdot N \cdot L \tag{1}$$

where *Abs* is the absorbance, $I_0(v)$ is the initial laser intensity obtained by the reference photodiode (PD1), $I(v)$ is the intensity of the laser after the absorbing obtained by the measurement photodiode (PD2), $v$ is the laser frequency, $I_{s0}$ and $I_s$ are the wavelength-independent signals originating from the detector dark current, *Tr* is the transmission of the laser with soot or other particles in the flame, $S(T)$ is the absorption line strength, $g(v - v_0)$ is the area normalized shape function, and $L$ is the optical path length. The detector dark current signal was obtained when the laser was blocked. In the clean, hot flue gases provided by the laminar flames, *Tr* was 1. In sooty environments, ln(*Tr*) was determined by spline interpolation of the deviation between the measured and the calculated $H_2O$ absorbance at five wavelengths (6484.07, 6484.59, 6484.93, 6485.42, and 6485.79 $cm^{-1}$). At these wavelengths, the absorption intensity is the lowest and the least sensitive to temperature. Several rounds of iterative calculation were performed to obtain *Tr*. At the beginning, an initially estimated concentration was used for *Tr* calculation. Based on the *Tr* value, the $H_2O$ concentration was determined through fitting between the calculated and measured water spectrum and used for another round of *Tr* value calculation.

## 3. Results and Discussion

The absorbance spectra over 6483.8–6485 $cm^{-1}$, obtained from the hot flue gas at 1260 K provided by flame F5, is shown in Figure 4a. It was attributed to the 8% $H_2O$ in the hot flue gas. Other major species, such as $CO_2$, were excluded due to their negligible absorption cross-section at this wavelength. The absorption spectrum of the 8% $H_2O$ at 1260 K was simulated using HITEMP2010 (Figure 4b). All water lines were recognized as hot lines. The spectral data, including transition wavenumber ($v$), line intensity ($S$) at 1260 K, lower-state energy ($E$), and the vibrational and rotational quantum numbers for the upper and lower states of typical transitions labeled a to f in Figure 4b (obtained from HITEMP2010) are summarized in Table 2. Compared with experimental results, the simulation can approximately predict strong lines, but the discrepancy is large, for instance, at the wavelength where the HCN (6484.78 $cm^{-1}$) and $C_2H_2$ (6484.03 $cm^{-1}$) lines are located.

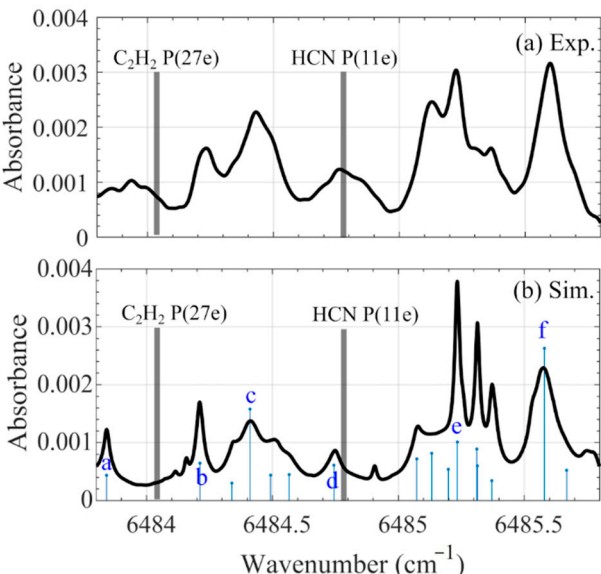

**Figure 4.** Absorbance spectra of $H_2O$ measured in the hot flue gas ($T$ = 1260 K) provided by flame F5 (**a**), and the calculated one of 8% $H_2O$ at 1260 K using HITEMP2010 (**b**). The chosen lines for HCN and $C_2H_2$ are marked with vertical bars.

**Table 2.** Spectra data of the six typical $H_2O$ transitions indicated in Figure 4b obtained from HITEMP2010. Transition wavenumber ($v$), line intensity ($S$) at 1260 K, lower-state energy ($E$), and the vibrational and rotational quantum numbers for the upper ($'$) and lower ($''$) states are presented.

| Tran. | $v$ (cm$^{-1}$) | $S$ (cm/Molecule) (1260 K) | $E$ (cm$^{-1}$) | $v'$ | $v''$ | $J'$ $Ka'$ $Kc'$ | $J''$ $Ka''$ $Kc''$ |
|---|---|---|---|---|---|---|---|
| a | 6483.840 | $2.91 \times 10^{-24}$ | 5076.349 | 2 0 1 | 1 0 0 | 9 1 9 | 10 3 8 |
| b | 6484.212 | $4.30 \times 10^{-24}$ | 6108.281 | 0 3 1 | 0 1 0 | 17 5 13 | 18 5 14 |
| c | 6484.411 | $1.05 \times 10^{-23}$ | 1282.919 | 0 2 1 | 0 0 0 | 8 1 7 | 9 3 6 |
| d | 6484.744 | $4.08 \times 10^{-24}$ | 4837.700 | 1 2 1 | 1 0 0 | 8 2 6 | 9 2 7 |
| e | 6485.234 | $6.73 \times 10^{-24}$ | 4738.634 | 0 2 1 | 0 0 0 | 19 3 17 | 20 3 18 |
| f | 6485.580 | $1.75 \times 10^{-23}$ | 3360.600 | 2 0 0 | 0 0 0 | 14 4 11 | 15 5 10 |

By measuring HCN and $C_2H_2$ in a combustion environment containing hot water vapor, a total absorbance contributed by HCN, $C_2H_2$, and $H_2O$ was obtained. To derive the concentrations of HCN and $C_2H_2$, the $H_2O$ absorption needs to be subtracted. As can be seen, the absorption spectrum resulting from the simulation using the HITEMP2010 database was not sufficiently accurate. Therefore, in the presented work, the temperature-dependent absorption spectra of hot water over 6483.8–6485 cm$^{-1}$ was experimentally measured. In the measurement, hot flue gas environments of 1120–1950 K were provided by the flames F1–F6 (cf. Table 1). Using the TDLAS system, the absorbance was measured under each flame condition. The measured $H_2O$ absorbance spectra are shown in Figure 5, where the absorbance was corrected to the value with an identical $H_2O$ number density, $4.7 \times 10^{17}$ molecule/cm$^3$ (10% at 1500 K). The absorbance data are available in the Supplementary Material. The water spectrum has a similar profile at different temperatures, but the absorption coefficient significantly increased with temperature.

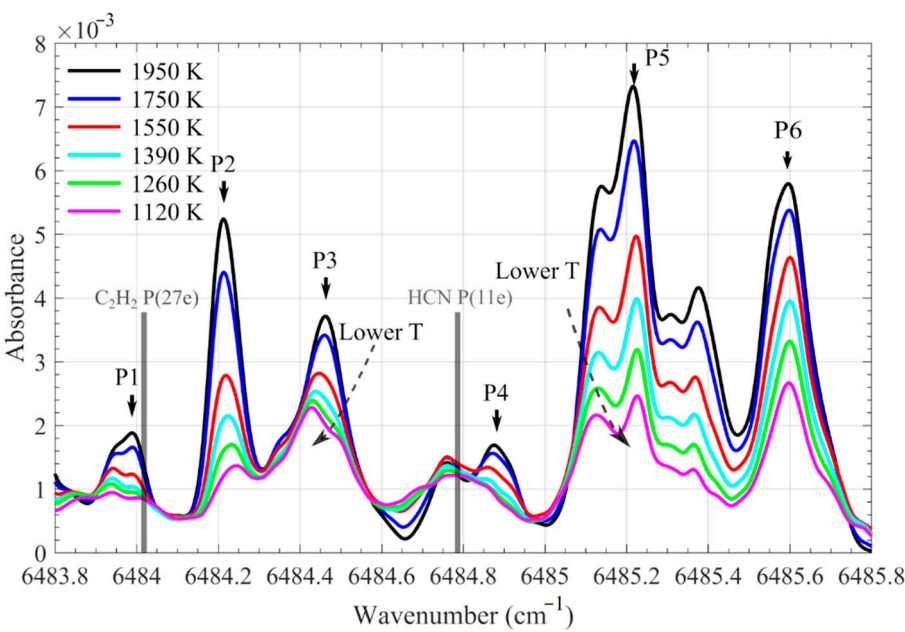

**Figure 5.** Measured $H_2O$ absorbance spectra between 6483.8 and 6485.8 $cm^{-1}$ at temperatures of 1120, 1260, 1390, 1550, 1750, and 1950 K. The absorbance was corrected to a value with an identical $H_2O$ number density, $4.7 \times 10^{17}$ molecules/$cm^3$ (10% at 1500 K). The typical six absorption peaks are labeled in the figure. The chosen lines for HCN and $C_2H_2$ are marked with vertical bars.

Six relatively strong hot water lines in the measured spectra are labeled at their respective peaks, P1–P6, as shown in Figure 5. The peak value as a function of temperature is plotted in Figure 6a. Nearly all linearly increased with temperature, but at different rates (e.g., the values at P2 and P3). Figure 5 shows that the relative intensity between the water lines with respective peaks at P2 and P3 changed significantly with temperature. The ratio of P3 to P2 as a function of temperature between 1000–2000 K is shown in Figure 6b, fitted by a quadratic equation, and expressed by:

$$R = 1.1505 \times 10^{-6} \times T^2 - 0.0047 \times T + 5.4049 \tag{2}$$

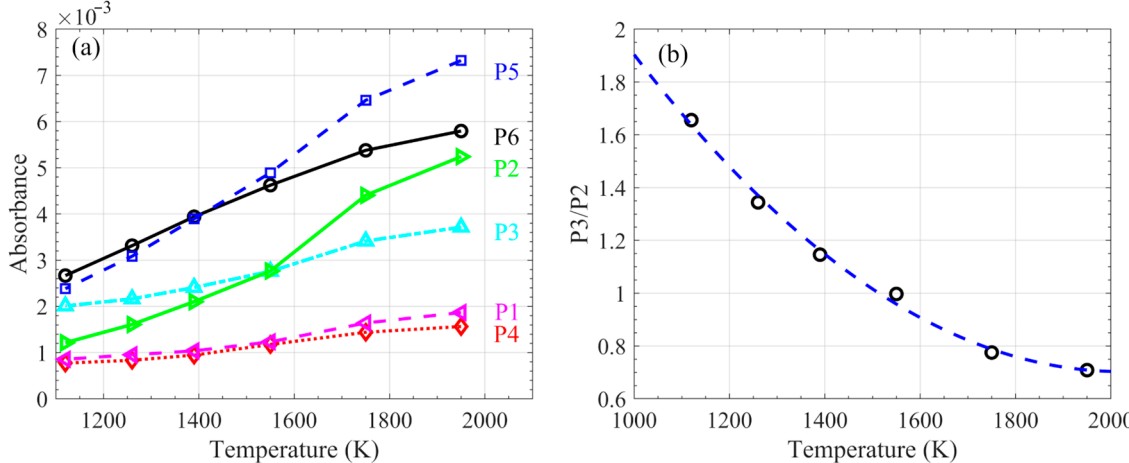

**Figure 6.** Absorbance of $H_2O$ at line peaks P1–P6 (as labeled in Figure 5) (**a**), and the ratio of P3 to P2 (**b**), as a function of temperature.

Therefore, the measured water absorption spectrum of these two lines between 6484.1 and 6484.6 $cm^{-1}$ can be used to evaluate the temperature. The uncertainty was estimated

to be approximately $\pm 6\%$ according to the equation fitting and the uncertainty regarding the TLAF measurement (about $\pm 3\%$). However, in harsh environments with uneven temperature distribution or soot particles, additional uncertainty will be introduced.

Since the absorption of HCN and $C_2H_2$ at 6484.78 and 6484.03 $cm^{-1}$ limited interference with these two water lines, in the HCN and $C_2H_2$ measurement, the gas temperature can be determined using the absorption profile of these two water lines, and based on the temperature, the absorption spectrum of the $H_2O$ mixed with HCN and $C_2H_2$ can be obtained through interception of the absorption spectra in Figure 5. Temperature can also be used to calculate the line intensity of HCN and $C_2H_2$, which is temperature-dependent.

The absorbance spectrum obtained in the plume 5 mm above burning nylon 66 strips in the hot flue gas environment (F7), at 1790 K with a residence time of 20 s, is shown in Figure 7 using black scattering circles. The absorption was attributed to $H_2O$, HCN, and $C_2H_2$. Using the aforementioned method, the $H_2O$ absorption between 6484.1 and 6484.6 $cm^{-1}$ was fitted through the interception of the absorption spectra at the same wavelength in Figure 5, with minimizing deviation. After the fitting process, the entire $H_2O$ absorption spectrum was obtained, i.e., the mega dot line in Figure 7. Based on Equation (2), the gas temperature was determined to be 1180 K. After subtracting the simulated $H_2O$ absorbance from the raw absorbance, the absorption spectrum of HCN near 6484.78 $cm^{-1}$ was obtained. By fitting using the parameters of the HCN line in the HITRAN2016 database and a Voigt line shape, represented by the blue dot line in Figure 7, the concentration of HCN was derived. The simulated $H_2O$ and HCN absorption was subtracted from the raw absorbance, and the remaining absorbance of near 6484.03 $cm^{-1}$, attributed to $C_2H_2$ absorption, was fitted using the parameters of the $C_2H_2$ line in the HITRAN2016 database and a Voigt line shape (Figure 7) to obtain the concentration. In this case, the concentration of HCN and $C_2H_2$ where the measurement uncertainty originated from the absorbance curve-fitting process was determined to be $820 \pm 190$ and $1170 \pm 340$ ppm, respectively. The large uncertainty in the spectrum fitting may originate from the uneven temperature of the gas plume and unrecognized species.

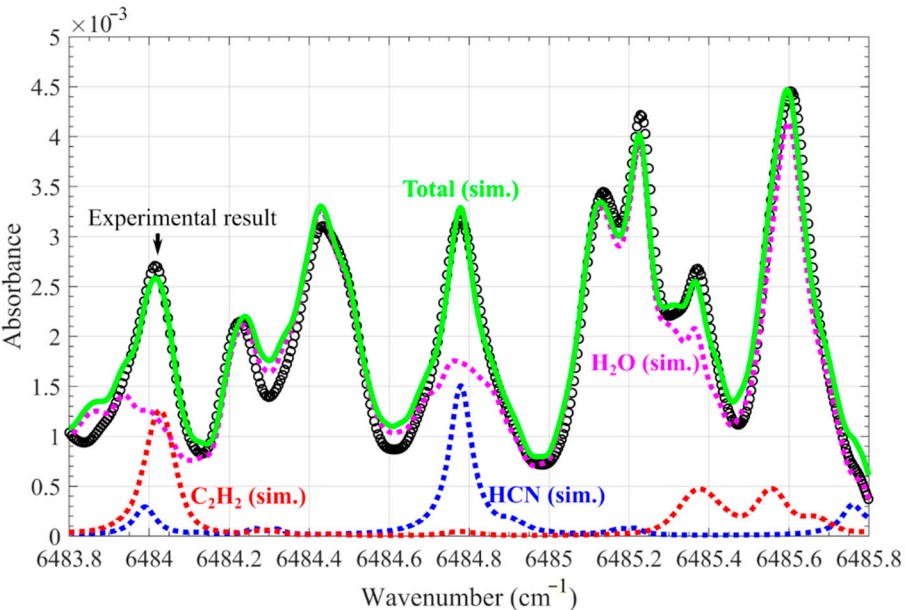

**Figure 7.** The absorbance spectrum obtained in the plume 5 mm above burning nylon 66 strips in the hot flue gas environment (F7), at 1790 K with a residence time of 20 s (black scattering circle) and the fitted one (green solid line), reflects the absorbance spectrum of 11% $H_2O$ at 1180 K obtained from our database (magenta dot line), along with the 820 ppm HCN (blue dot line) and 1170 ppm $C_2H_2$ (red dot line) using the HITRAN2016 database.

The measured concentrations of HCN and $C_2H_2$ in the plume of the burning nylon 66 strips as a function of residence time are shown in Figure 8. After the first 16 s of preheating, the release of HCN and $C_2H_2$ almost simultaneously began, and lasted over 20 s. The maximum concentration of HCN was determined to be 4000 ± 820 ppm at 26 s, and the maximum concentration of $C_2H_2$ was determined to be 3800 ± 480 ppm at 22 s. During the gas release process, the temperature dropped from about 1220 to 1000 K, similar to the observation of the volatile release of burning biomass particles [2,26].

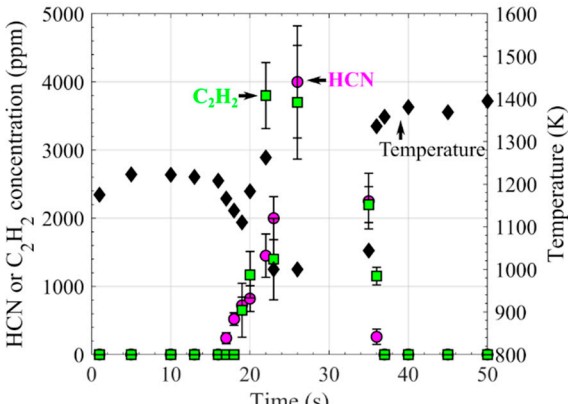

**Figure 8.** Variation in the HCN and $C_2H_2$ concentrations in the plume 5 mm above burning nylon 66 strips in the hot flue gas at 1790 K provided by flame F7, and the local temperature obtained based on Equation (2). (HCN: magenta circle, $C_2H_2$: green square, temperature: black diamond).

## 4. Conclusions

Quantitative and simultaneous measurement of HCN and $C_2H_2$ in a combustion environment, using TDLAS at around 6484 cm$^{-1}$, was developed in the presented work. The P(11e) line of the first overtone vibrational band $v_1$ of HCN at 6484.78 cm$^{-1}$ and the P(27e) line of the $v_1 + v_3$ combination band of $C_2H_2$ at 6484.03 cm$^{-1}$ were selected. It was discovered that the main challenge of accurately measuring HCN and $C_2H_2$ was the interference of water vapor absorption in the combustion environment. To eliminate the influence, accurate knowledge of the hot water lines was important. However, the simulation using the most prevailing databases, HITRAN2016 and HITEMP2010, was inconsistent with the experimental measurement. Thus, an important step was carrying out a comprehensive investigation of the temperature-dependent water absorption in various hot gas environments at temperatures ranging from 1120 to 1950 K. Based on the measured absorption spectra of $H_2O$ at different temperatures, the HCN and $C_2H_2$ absorption spectra were resolved after $H_2O$ absorption subtraction, and the concentrations of HCN and $C_2H_2$ were calculated through the spectral fitting using the HITRAN database. The technique was applied to simultaneously measure the temporally resolved release of HCN and $C_2H_2$ over burning nylon 66 strips in a hot oxidizing environment at 1790 K. In the hot plume 5 mm above the stripes, the maximum concentration of HCN was detected to be 4000 ± 820 ppm at the residence time of 26 s, and $C_2H_2$ was detected to be 3800 ± 480 ppm at 22 s.

**Supplementary Materials:** The following are available online at https://www.mdpi.com/article/10.3390/pr9112033/s1, The supplementary material includes the measured hot water absorption spectra between 6483.8 and 6485.8 cm$^{-1}$ at 1120, 1260, 1350, 1550, 1750, and 1950 K.

**Author Contributions:** Conceptualization, W.W. and Z.L.; methodology, W.W. and Z.L.; investigation, W.W.; resources, W.W.; data curation, W.W.; writing—original draft preparation, W.W.; writing—review and editing, Z.L.; visualization, W.W.; project administration, Z.L.; funding acquisition, M.A. All authors have read and agreed to the published version of the manuscript.

**Funding:** The work was financially supported by the Swedish Energy Agency (KC-CECOST, 22538-4 biomass project), the Knut & Alice Wallenberg Foundation (COCALD KAW2019.0084), European Research Council ERC Advanced Grant TUCLA 669466) and the Swedish Research Council (VR).

**Institutional Review Board Statement:** Not applicable.

**Informed Consent Statement:** Not applicable.

**Data Availability Statement:** Not applicable.

**Conflicts of Interest:** The authors declare no conflict of interest.

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
