# Peer review of "Simultaneous Quantitative Detection of HCN and C2H2 in Combustion Environment Using TDLAS"

_processes, doi:10.3390/pr9112033_

Round 1
Reviewer 1 Report
The paper entitled « simultaneous quantitative detection of HCN and C2H2 in combustion environment using TDLAS » presents not only experimental measurements of HCN and C2H2 but a large section is also dedicated to the quantitative measurement of water vapor. Indeed in the IR range, HCN and C2H2 absorption lines are embedded with the extended absorption of H2O.
To prevent this the authors have thoroughly 1/selected 2 individual lines for HCN and C2H2 whose line strengths are perfectly reported in HITRAN database, respectively at 6484.78 and 6484.03 cm-1, and 2/ considered the contribution of the hot water to the experimental absorption spectrum. Regarding the H2O treatment, I have few important issues. This is why I am asking for major revision before any publication in the journal “Processes”.
- Introduction,
The authors present well the state-of-the art for quantitatively measuring HCN and C2H2 in the IR wavelength range. Figure 1 illustrates the interferences between these 2 species and also the difficulties encountered with the hot water vapor in combustion. Please note that HCN was measured with a pulsed CRDS in [5], and by cw-CRDS later in [6].
- Experimental setup
Why the synthetic absorption spectra are shown for 1-m optical path length when the experimental optical path is only 34 cm?
Equation 1, How is determined Tr. You indicate it is 1 in non –sooting flame, determined from a curve fitting in sooty environment. But was is this curve fitting? Please clarify this point.
How is measured the
To by pass the difficulties due to the hot water, the authors chose to experimentally determine its absolute contribution to the experimental absorption spectra measured in the hot flus gas of various flames. For that, they assume a mole fraction of 8% in the hot flue gas without any explanations. Moreover, the same mole fraction seems to be assumed in all the examined flames while they are stabilized at various equivalence ratio, different temperature of the gas product. Without clear explanation on the calculation / measure of the water concentration, it seems impossible to extract the absorption cross section like this. From an absorbance signal, you can determine the concentration knowing the absorption cross section, or the absorption cross section knowing the concentration. Here, there is no indication on how is the concentration is determine. It appears that the so-called absorption cross section is erroneously named since there is no evidence that the mole fraction of the hot water is independent of the flame. Please this section and the discussion later has to re-examined.
The labels of the transition are (a-f) in Table 2 and Figure 4, but P1-P6 later (fig. 5). Please rename them.
You present that the temperature can be determine from the ratio beween the line intensities P2/P3 which is effectively independent of the knowledge of the mole fraction of H2O. Can you indicate the uncertainties of the temperature according to Eq. 2. In the following, did you consider the temperature measured using TLAF or the ones determined from Eq. 2?
Figure 5 and Figure 6 definitively present the absorbance or absorption coefficient, not the absorption cross section.
Reviewer 2 Report
Authors presented an interesting methodology to determine simultaneously the concentration of C2H2 and HCN at in combustion environment. Moreover authors determine the temperature of the gas mixture by calculating the ratio of the peaks due to 2 ro-vibrational lines of H2O. Here according to authors the sensor is based on a TDLAS which was implemented by using a DBF laser. Moreover, authors presented experimental measurements to support their simulations and these let us to appreciate that these have a good agreement. Therefore, I would like to recommend to consider the manuscript for publication in the journal after some moderate corrections are performed.
Some points that I consider should be attended are:
1- I would like to suggest to include a brief description about the Tunable Laser Spectroscopy technique. This for clarity purposes thinking in readers that are not experts in TLAS. Moreover, and again for clarity purposes explain the advantages for splitting the source signal into the measurement and the reference channel. Here if possible I would like to suggest include the following references:
a) Portable Tunable Diode Laser Absorption Spectroscopy System for Dissolved CO2 Detection Using a High-Efficiency Headspace Equilibrator. Sensors 2021, 21, 1723. https://doi.org/10.3390/s21051723
b) Tailored Algorithm for Sensitivity Enhancement of Gas Concentration Sensors Based on Tunable Laser Absorption Spectroscopy. Sensors 2018, 18, 1808. https://doi.org/10.3390/s18061808
c) Recent Developments in Modulation Spectroscopy for Methane Detection Based on Tunable Diode Laser. Appl. Sci. 2019, 9, 2816. https://doi.org/10.3390/app9142816
2- After equation (1), please describe, in a detailed way, the procedure used to calculate Tr.
3- In figure 7, it is presented the absorbance spectrum obtained in the plume 5 mm. Here as far as can be understood, if the simulated spectra of H2O, C2H2 and HCN are ‘combined’ the experimental spectra can be obtained. In this point I would like to suggest to provide a detailed explanation (steps) about the procedure used to determine, from the simulated spectrum, the concentrations of C2H2 and HCN. I consider that this discussion is needed since basically it is one of the most important part for implementing and measuring with the real sensor/system.
Reviewer 3 Report
This paper reports simultaneous detection of HCN and C2H2 in combustion environment based on TDLAS. To alleviate the interference of water at high temperature the authors have used spectroscopic data obtained by themselves instead of common database of HITRAN and HITEMP for water. This was because HITRAN and HITEMP databases present different water absorption spectra.
- Figs. 4(a) and 4(b) show the discrepancy between the absorbance spectra of water measured by the authors and HITEMP database. I would like to hear from the authors about the cause of the difference among the authors’ spectroscopic data, HITRAN and HITEMP data for hot water, including the spectrum resolution of the authors’ TDLAS system.
- The direction of the vertical axis of Fig. 4(b) should be the same with that of Fig. 4(a) for the convenience of readers.
Round 2
Reviewer 1 Report
The authors have considered my previous remarks.
First Tr in sooty environment is determined from a minimization between the experimental and simulated absorbance of H2O, focusing on 5 transitions selected because of their weak intensity and low temperature dependency. However, in such environment how to estimate the H2O concentration?
The determination of an absorption cross section requires the knowledge of the absorbance of the selected species AND its concentration. Indeed, the authors now specify that the concentration of H2O is not constant whatever the conditions of the flame. Here, the concentration is only an estimated value “based on equilibrium chemical calculation”. This explanation is rather poor. How do you simulate this type of flame? You put 100% confidence in the calculated concentration and then seem to forget that it is just an estimated value, as for example in the simulated spectra shown in Fig. 5.
I am very bothered by the fact that you have plotted the “absorption cross section” (based on an assumption of the H2O concentration), not the absorbance. What you want to determine is the concentrations of HCN and C2H2. From the absorbance spectrum you want to substract the H2O contribution. So, I don't understand why you insist to plot the absorption cross section. In the end, you will get exactly the same data in Fig. 8, which is exactly the point of your paper.
Round 3
Reviewer 1 Report
Thanks for having considered my comments.